# Meibomian Gland Dysfunction Is Associated with Low Levels of Immunoglobulin Chains and Cystatin-SN

**DOI:** 10.3390/ijms242015115

**Published:** 2023-10-12

**Authors:** Danson Vasanthan Muttuvelu, Lasse Jørgensen Cehofski, Jeppe Holtz, Tor Paaske Utheim, Xiangjun Chen, Henrik Vorum, Steffen Heegaard, Marie Louise Roed Rasmussen, Asif Manzoor Khan, Ahmed Basim Abduljabar, Bent Honoré

**Affiliations:** 1Faculty of Health and Medical Sciences, University of Copenhagen, 2200 Copenhagen, Denmark; dansonvm@gmail.com; 2Department of Ophthalmology, Odense University Hospital, 5000 Odense, Denmark; jeppe.holtz@rsyd.dk; 3Department of Clinical Research, University of Southern Denmark, 5000 Odense, Denmark; 4Department of Medical Biochemistry, Oslo University Hospital, 5000 Oslo, Norway; utheim2@gmail.com (T.P.U.); chenxiangjun1101@gmail.com (X.C.); 5Norwegian Dry Eye Clinic, 0366 Oslo, Norway; 6Department of Ophthalmology, Aalborg University Hospital, 9000 Aalborg, Denmark; henrik.vorum@rn.dk; 7Department of Clinical Medicine, Aalborg University, 9000 Aalborg, Denmark; 8Department of Ophthalmology, Rigshospitalet, University of Copenhagen, 2100 Copenhagen, Denmark; sthe@sund.ku.dk (S.H.);; 9Department of Biomedicine, Aarhus University, 8000 Aarhus, Denmark; amanzoor@biomed.au.dk (A.M.K.); ahmed@biomed.au.dk (A.B.A.)

**Keywords:** blepharitis, tear, dry eye, meibomian gland dysfunction, mass spectrometry, proteome, proteomics, immunoglobulin, cystatin

## Abstract

Meibomian gland dysfunction (MGD) is a highly prevalent condition and the most common cause of evaporative dry eye disease. Studying the proteome of MGD can result in important advances in the management of the condition. Here, we collected tear film samples from treatment naïve patients with MGD (n = 10) and age-matched controls (n = 11) with Schirmer filtration paper. The samples were analyzed with label-free quantification nano liquid chromatography—tandem mass spectrometry. The proteins were considered differentially expressed if *p* < 0.05. A total of 88 proteins were significantly regulated. The largest change was observed in cystatin-SN, which was downregulated in MGD and correlated negatively with tear meniscus height. The downregulation of cystatin-SN was confirmed with targeted mass spectrometry by single reaction monitoring (SRM). Eighteen immunoglobulin components involved in B cell activation, phagocytosis, and complement activation were downregulated in MGD including Ig alpha-1 chain C region, immunoglobulin J chain, immunoglobulin heavy variable 3–15, and Ig mu chain C region. The changes in cystatin-SN and immunoglobulin chains are likely to result from the inflammatory changes related to tear film evaporation, and future studies may assess their association with the meibum quality.

## 1. Introduction

Dry eye disease (DED) is generally associated with a significant reduction in physical and mental quality of life [1]. DED may be broadly subdivided into aqueous-deficient DED (reduced lacrimal secretion), evaporative eye disease (excessive tear film evaporation) or a combination of the two [2]. Meibomian gland dysfunction (MGD) is the most important cause of evaporative dry eye disease [3]. MGD is highly prevalent, with more than 85% of DED cases demonstrating signs of MGD [2]. MGD is a type of posterior blepharitis localized posterior to the grey line of the eyelid characterized by reduced meibum secretion and/or changes in the meibum composition leading to increased tear film evaporation [3]. MGD often involves terminal duct obstruction with plugging of the meibomian orifices [4]. The increased tear film evaporation triggers a vicious cycle of tear film instability, desiccation, inflammation, and apoptosis of cells in the ocular surface [2].

The conservative management of MGD includes warm compresses and lid hygiene, but medical treatments are emerging, including antibiotics, nonsteroidal and steroidal anti-inflammatory agents, fatty acid supplementation, and Demodex infestation [3]. Antibiotics such as tetracyclines and azithromycin have been found to reduce signs of eyelid inflammation. Oral supplementation with omega-3 fatty acids has been found to be associated with improved tear break-up time and Schirmer test scores [3]. 

While changes in the lipid composition of the tear film is a well-established component in the etiology of MGD [5], systematic protein studies of MGD with proteomic techniques remain limited. The overexpression of annexin-A1, clusterin, alpha-1-acid glycoprotein 1, and lactoperoxidase has previously been reported in MGD, while the downregulation of thioredoxin-1, Ig gamma-1, membrane-associated phospholipase A2, and antileukoproteinase has been reported. These protein changes were suggested to be driving forces of inflammation and oxidative stress in MGD [6]. 

The identification of tear film proteins associated with MGD can lead to significant therapeutic advances. The objective of proteome studies is to identify and quantify the entire set of proteins in a tissue or body fluid. With significant advances in sample processing, liquid chromatography, and mass spectrometry, modern proteomic techniques allow for in-depth large-scale protein analysis of the tear film [7,8,9]. Here, we compared tear film samples from patients with MGD with an age-matched control group using advanced proteomic analysis through label-free quantification. 

## 2. Results

### 2.1. Significantly Regulated Proteins

A total of 2407 proteins were successfully identified after filtering (Appendix A). In total, 927 proteins were successfully identified and quantified in ≥70% of the samples in each of the two groups (Appendix A), and 88 proteins were significantly differentially expressed in MGD (Table 1) (Figure 1). Among the significantly regulated proteins, 58 proteins were significantly upregulated, while 30 proteins were significantly downregulated (Table 1). 

The largest downregulations were observed in the cystatin-SN (fold change = 0.23), apolipoprotein D (fold change = 0.30), and Ig mu chain C region (fold change = 0.46) (Table 1). The strong downregulation of cystatin-SN was also analyzed using targeted mass spectrometry using SRM (Figure 2). The median amount of cystatin-SN in the blepharitis group was downregulated with a fold-change of 0.26 (*p* = 0.0079) (Figure 2), in very good agreement with the fold-change observed by discovery-based proteomics. Cystatin-SN correlated negatively with tear meniscus height (Figure 3), while cystatin-SN did not correlate with the Schirmer test, OSDI, and NBUT (*p* > 0.05). The proteins with the largest increases in MGD included polyadenylate-binding protein 1 (fold change = 2.67), dynactin subunit 1 (fold change = 2.50), and exportin-2 (fold change = 2.27).

Seven proteins correlated with NBUT. The proteins with positive correlations with NBUT included myosin regulatory light chain 12B, ubiquitin carboxyal-terminal hydrolase 47, hsc70-interacting proteins, and poly(rC)-binding protein 2. Proteins with negative correlations with NBUT included endoplasmic reticulum aminopeptidase 1, immunoglobulin heavy variable 3–15, and Ig mu chain C region (Table 2). Myosin-14 was found to correlate with the Schirmer test (r = 0.64; *p* = 0.047).

### 2.2. Bioinformatics

MGD was associated with the regulation of proteins involved in the B cell receptor signaling pathway, the positive regulation of B cell activation, phagocytosis, and complement activation (Figure 4A). The group of proteins involved in B cell receptor signaling, phagocytosis, and complement activation consisted of a large number of immunoglobulin components, including immunoglobulin lambda constant 3, Ig mu chain C region, Ig kappa chain C region, Ig alpha-1 chain C region, immunoglobulin heavy variable 5–51, immunoglobulin heavy variable 3–15, immunoglobulin heavy variable 3–49, immunoglobulin heavy variable 3–72, Ig alpha-1 chain C region, and Ig kappa chain C region (Figure 4B). All immunoglobulin components were downregulated in MGD (Table 1). STRING cluster analysis identified a cluster of immunoglobulin heavy variable 3–15, immunoglobulin heavy variable 3–43D, and immunoglobulin J chain (Figure 5). MGD was also associated with the regulation of proteins involved in proteasomal catabolic processes, including proteasome subunits alpha types 1, 5, and 7, and proteasome subunit beta type 1 (Figure 4A,B). All proteasome subunits were upregulated (Table 1).

Proteasome subunits also formed the largest cluster of the dataset (Figure 5), which consisted of proteasome subunits, 26S protease regulatory subunit 8, 26S proteasome non-ATPase regulatory subunit 1, 26S proteasome non-ATPase regulatory subunit 5, and proteasome subunit beta type 1. The cluster of proteasome subunits also consisted of t-complex protein 1 subunit theta, t-complex protein 1 subunit beta, and t-complex protein 1 subunit zeta (Figure 5).

The regulation of the proteins involved in deubiquitination was observed in MGD, including proteasome subunits and ubiquitin carboxyl-terminal hydrolase 14, ubiquitin carboxyl-terminal hydrolase 47, and ubiquitin carboxyl-terminal hydrolase isozyme L3 (Figure 4A,B). All ubiquitin carboxy-terminal hydrolases were upregulated in MGD (Table 1). The STRING cluster analysis showed similar results with the identification of a major cluster consisting of ubiquitin-like modifier-activating enzyme 6, ubiquitin-like modifier-activating enzyme 7, ubiquitin-conjugating enzyme E2 variant 1, ubiquitin-conjugating enzyme E2 N, ubiquitin carboxyl-terminal hydrolase isozyme L3, and ubiquitin carboxyl-terminal hydrolase 47 (Figure 5).

## 3. Discussion

Our study reports on tear film proteome changes in patients with MGD defined as MGD. The most pronounced change was observed in cystatin-SN. The downregulation of cystatin-SN was confirmed by targeted mass spectrometry, thereby confirming findings obtained with discovery-based proteomics. The downregulation of cystatin-SN has previously been reported in tear fluid samples from patients with fungal keratitis and ocular graft versus host disease [10,11]. Cystatin proteins have a protective function regulating endogenous cysteine proteases which may cause uncontrolled proteolysis and tissue damages if inadequately regulated [12]. As cystatin-SN exerts antimicrobial activity, a reduced tear film level of cystatin-SN has been hypothesized to be associated with an increased risk of infection [13,14]. While a number of studies have identified decreased levels of cystatin-SN, there is a need for future studies to assess the association between cystatin-SN and the quality of the meibum.

Our study identified an extensive downregulation of 18 immunoglobulin components. There is only limited knowledge about the role of immunoglobulins in blepharitis, but our data indicate that immunoglobulins may play a major role in the molecular processes underlying the condition. Changes in immunoglobulins are likely to be involved in the inflammatory changes resulting from increased evaporation in MGD. Immunoglobulin heavy variable 3–15 and Ig mu chain C region correlated negatively with NBUT, indicating that changes in immunoglobulin components affect the quality of the tear film. Drew et al. [15] pointed out that a limitation in the use of artificial tears is the lack of components of natural tears, including immunoglobulins. The processes driven by immunoglobulins in MGD warrant further investigation, as changes related to immunoglobulins are likely to affect the homeostasis of the ocular surface.

Myosin regulatory light chain 12B was found to correlate with NBUT while myosin 14 correlated with Schirmer test, measured in mm of wetting of the strip. Myosin regulatory light chains are also expressed in non-muscle tissue and serve to maintain cellular integrity [16]. Myosin 14 is a non-muscle myosin involved in cytoskeletal rearrangement and organelle translocation [17]. Thus, myosin regulatory light chain and myosin 14 are likely to contribute to the homeostasis of the ocular surface through structural changes.

Correlations with NBUT were observed for ubiquitin carboxyl-terminal hydrolase 47, hsc70-interacting protein, and poly(rC)-binding protein 2, but the roles of these proteins at the ocular surface level remain unknown. Ubiquitin carboxyl-terminal hydrolase 47 is a ubiquitin-specific protease that regulates cell survival and a regulator of base excision repair [18,19]. Poly(rC)-binding protein 2 is an RNA-binding protein involved in pre-mRNA splicing, mRNA splicing, mRNA stabilization, and translational control [20]. Hsc70-interaction protein is a chaperone which binds to unfolded proteins and prevents them from refolding [21]. 

A bioinformatic analysis of regulated proteins identified an upregulation of proteins involved in the ubiquitin-proteasome system, which degrades short-lived and regulatory proteins as well as damaged and misfolded proteins [22]. The proteins targeted for proteasomal degradation are conjugated with ubiquitin, whereby ubiquitylation serves to control the ubiquitin–proteasome system [22]. Blepharitis has previously been reported as an ocular complication in patients with multiple myeloma treated with the proteasome inhibitor bortezomib [23], suggesting an association between proteasomal regulation and blepharitis. Changes to the ubiquitin-proteasome system are likely to cause changes in protein degradation which affect the tear film evaporation. However, further studies are required to establish the role of the ubiquitin–proteasome system.

## 4. Materials and Methods

### 4.1. Samples

The study was approved by the Regional Committees for Medical Health and Research Ethics (Permission ID: 134689) and adhered to the tenets of the Helsinki Declaration. Subjects with MGD and a control group were recruited from the Norwegian Dry Eye Clinic, Oslo, Norway. If the patients were found to meet the criteria of the study, the nature of the study was carefully explained, and written information about the study was provided. Informed written consent was obtained from all included subjects prior to their study visit. Patients with posterior blepharitis defined as MGD (n = 10), and age-matched controls without any history of ocular disease (n = 11), were recruited for the study (Table 3). The controls are a group of healthy subjects we previously reported on [24]. Statistical analysis using Student’s *t*-test was used to verify that there was no statistically significant difference in age between the two groups. Statistically significant differences between the MGD group and the control group in terms of Schirmer test, OSDI, tear meniscus height, and NBUT were verified using Student’s *t*-test (Table 3). 

The inclusion criteria were ≥18 years of age by the time of sample collection, and patients in the MGD group were required to have clinical findings consistent with MGD. MGD was diagnosed based on the recommended definition published by the International Workshop on MGD [25], which classifies MGD as a chronic, diffuse abnormality of the meibomian glands, commonly characterized by terminal duct obstruction and/or qualitative/quantitative changes in the glandular secretion. 

The exclusion criteria were the following: withdrawn consent, blepharitis secondary to other ocular condition or dermatological disease, concomitant topical or intraocular treatment, use of topical or systemic corticosteroids less than 30 days prior to sample collection, use of topical or systemic antibiotics less than 30 days prior to sample collection, use of warm compresses less than 30 days prior to sample collection, any active intraocular inflammation, any history of uveitis in either eye, or previous exposure to therapeutic radiation in the periorbital region.

All subjects completed the Ocular Surface Disease Index Questionaire (OSDI) [26] and underwent a full ophthalmological examination. An Oculus Keratograph 5M (Oculus Optikgeräte, Wetzlar, Germany) was used to perform meibography and to measure the tear meniscus height and non-contact break-up time (NBUT). The meiboscore was assessed from infrared images of the lower eyelid with th Ocular Keratograph 5M and grading was performed as previously described [27]. Grade 0 represented no loss of meibomian glands, grade 1; <1/3 of glands lost, grade 2; 1/3–2/3 glands lost, and grade 3; >2/3 glands lost. NBUT was measured as previously described [27,28]. For the assessment of NBUT, patients were instructed to blink twice and then refrain from blinking for as long as possible, but maximally for 24 s. After the second blink, the measurement started automatically and tear film disruption was detected automatically using the Oculus Keratography 5M. NBUT was assessed three times and the average value was applied.

Tear film samples were obtained using a Schirmer’s tear test strip (HAAG-STREIT, Essex, UK) as previously described [29]. Briefly, the test strip was placed on each eye for five minutes or more until at least 10 mm tear volume was obtained. Test strips were stored at −80 °C in Eppendorf tubes until further use.

### 4.2. Sample Preparation for Mass Spectrometry

The samples were stored at −80 °C until preparation was initiated. By the beginning of the preparation, the samples were thawed and Schirmer strips incubated for 20 min in lysis buffer (5% SDS, 50 mM triethylammonium bicarbonate [TEAB]). Samples were prepared according to the S-Trap™ Micro spin column digestion protocol from ProtiFi (ProtiFi, Huntington, NY, USA) as previously described [8].

The reduction in disulphide bonds, alkylation of cysteines, and digestion in S-Trap micro columns were performed as described in a recent article [8]. The elution of peptides, recovery of hydrophobic peptides, and measurement of peptide concentration were performed as previously described [8,30]. Each sample was dried in a vacuum centrifuge and stored at −80 °C until further use.

### 4.3. Label-Free Quantification Nano Liquid Chromatography–Tandem Mass Spectrometry

Samples from patients with MGD were compared to control samples using label-free quantification nano liquid chromatography tandem mass spectrometry (LFQ nLC-MS/MS). The samples were analyzed in replicates, except for a few samples which were analyzed in triplicates. Samples were re-suspended in 0.1% formic acid followed by LFQ nLC-MS/MS. Of each sample, 1.0 µg was analyzed. LFQ nLC-MS/MS was performed on an Orbitrap Fusion Tribrid mass spectrometer (Thermo Fisher Scientific Instruments, Waltham, MA, USA) coupled to a Dionex UltiMate^TM^ 3000 RSLC nano system and an EasySpray™ ion source (Thermo Fisher Scientific Instruments, Waltham, MA, USA). Liquid chromatography and label-free quantification were performed as previously reported [8] with a few modifications. The orbitrap scan range (*m/z*) was 375–1500. The elution gradient was 3 h, established by mixing buffer A (99.9% water and 0.1% formic acid) and buffer B (99.9% acetonitrile and 0.1% formic acid). Using the MaxQuant software version 1.6.6.0 (Max Planck Institute of Biochemistry, Martinsried, Germany; https://maxquant.net/maxquant) for label-free quantification (LFQ) analysis [31], raw data files were searched against the UniProt *Homo sapiens* database using settings as previously defined [32]. The unfiltered results of the database search are available in Appendix A. 

Mass spectrometry data were further processed with Perseus software version 1.6.2.3 (Max Planck Institute of Biochemistry, Martinsried, Germany; https://maxquant.net/perseus) to remove poorly identified proteins as described in a previous article [33]. LFQ values were log_2_ transformed and mean LFQ values were calculated. At least two unique peptides were required for successful protein identification. Successful identification and quantification were required in at least 70% of the samples in each group.

### 4.4. Statistics

Statistical analysis by Student’s t-test was performed on proteins that were successfully identified and quantified in at least 70% of the samples in each group. Proteins were considered differentially expressed if *p* < 0.05. Volcano plots were created with STATA 16.0 (StataCorp, College Station, TX, USA). Correlations between key proteins selected for validation and clinical parameters were calculated with Pearson’s correlation coefficient (r) in STATA 16.0. Correlations were considered statistically significant if *p* < 0.05. Two-way scatter plots with prediction from a linear regression were created in STATA 16.0.

Bioinformatic analyses of statistically, significantly regulated proteins were performed using GeneCodis 4 software [34] (genecodis.genyo.es) as described in a previous paper [35]. Cluster analysis was performed in STRING 11.5 (string-db.org) [36,37,38] as reported in a recent article [8], with the minimum required interaction score set to 0.4 and the Markov Cluster Algorithm set to 3.

### 4.5. Targeted Mass Spectrometry with Single Reaction Monitoring (SRM)

Single reaction monitoring (SRM) was performed using a TSQ Quantiva mass spectrometer (Thermo Fisher Scientific Instruments, Waltham, MA, USA) as previously described [24]. The data analysis was carried out using Skyline v22.1 [39] (ref: Pino LK et al. 2020) and further calculations were performed in Excel. A standard peptide for cystatin-SN was labelled with the heavy isotopes ^13^C and ^15^N at the C terminal R (^13^C_6_,^15^N_4_) (ThermoFisher Scientific, standard peptide, purity >50%), IIPGGIYNADLNDEWVQR, [M + 2H]^2+^ = 2084.0556 Da. The lyophillized peptide was dissolved in 0.1% formic acid at 1 µg/µL. For each sample, 1 µg of generated tryptic peptides was injected together with 0.1 ng (48.04 × 10^−15^ mole) of the heavy cystatin-SN peptide. Samples were injected in duplicate. The heavy peptide peak area and the corresponding light peptide peak area were compared to calculate the molar amount of light peptide in each sample. By assuming the complete tryptic digestion of proteins in the samples, the molar amount of cystatin-SN was considered to be equal to the amount of light peptide generated. The amount of cystatin-SN protein per gram of total peptide in the samples was estimated by using the deduced molecular mass of cystatin-SN from Uniprot 16,388 Da. The technical duplicates had an average coefficient of variation at 2.3%.

## 5. Conclusions

In conclusion, the proteomic analysis suggested a multifactorial pathogenesis in MGD. The downregulation of cystatin-SN and immunoglobulins is likely to be associated with alterations to the ocular surface immune response following increased evaporation. The decreased level of cystatin-SN is likely to render patients with blepharitis more susceptible to infection as cystatin-SN has antimicrobial features. MGD was associated with changes in the proteins involved in protein degradation, which may play a central role in the processes leading to increased evaporation.

## Figures and Tables

**Figure 1 ijms-24-15115-f001:**
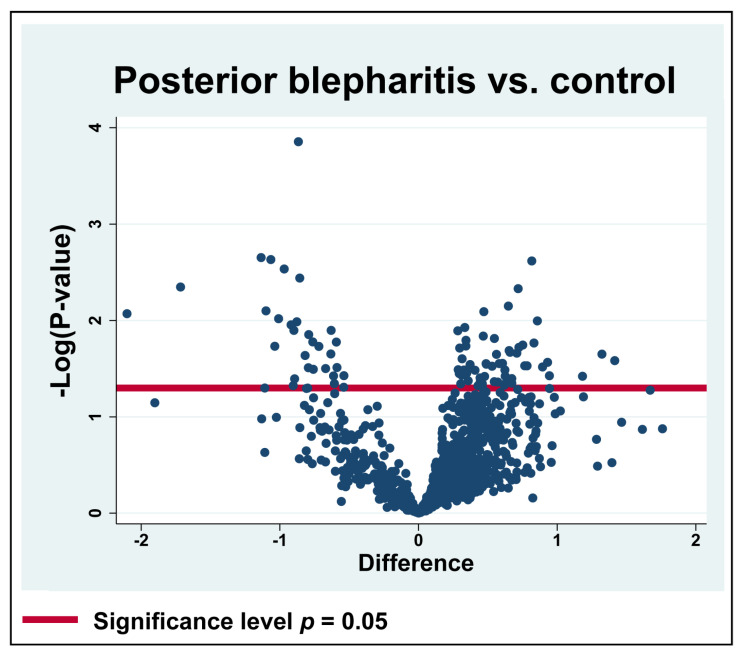
Volcano plot. Log_2_ transformed abundance ratios (MGD/control) for each protein are plotted on the *x*-axis. Negative log_10_ transformed *p*-values are plotted on the *y*-axis. Statistically significantly changed proteins are localized above the red horizontal line, which indicates the significance level of *p* = 0.05. Among the significantly regulated proteins, 58 proteins were increased in content, while 30 proteins were decreased in content.

**Figure 2 ijms-24-15115-f002:**
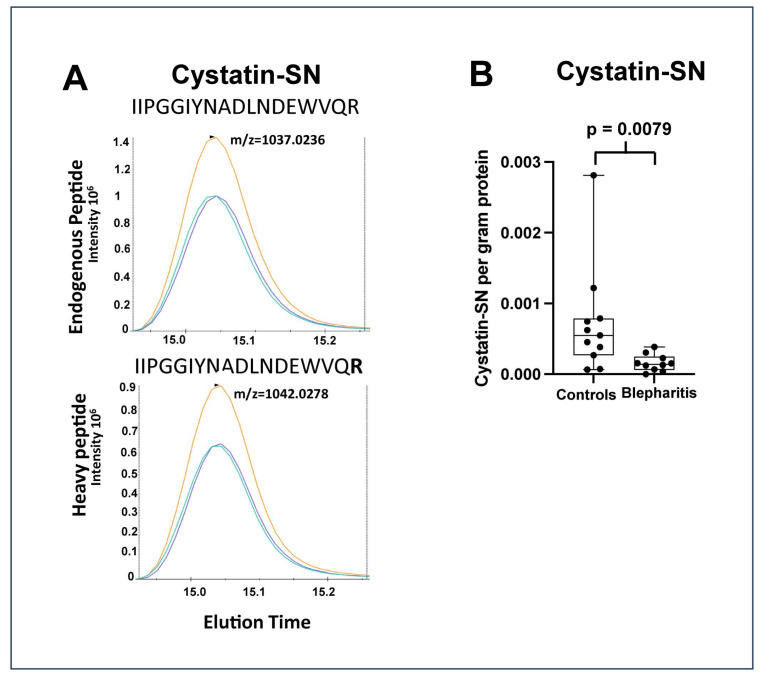
(**A**) Chromatograms of the peaks obtained by single reaction monitoring analysis of cystatin-SN with added heavy peptide, IIPGGIYNADLNDEWVQR, synthesized with the C-terminal R (bold) containing heavy isotopes (^13^C_6_,^15^N_4_). Three transitions are shown with different colours from the endogenous peptide at *m*/*z* = 1037.0236 (upper panel) and from the heavy peptide at *m*/*z* = 1042.0278 (lower panel). (**B**) Box plot of the amount of cystatin-SN (gram) per gram protein in the healthy and the blepharitis group. Cystatin-SN was downregulated 0.26-fold in the blepharitis group (Mann–Whitney test, *p* = 0.0079).

**Figure 3 ijms-24-15115-f003:**
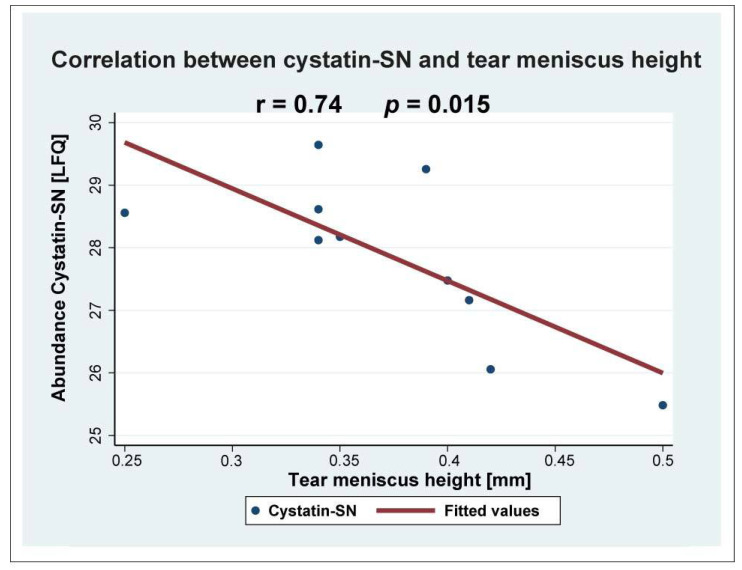
Cystatin-SN correlated negatively with tear meniscus height (r = 0.74; *p* = 0.015).

**Figure 4 ijms-24-15115-f004:**
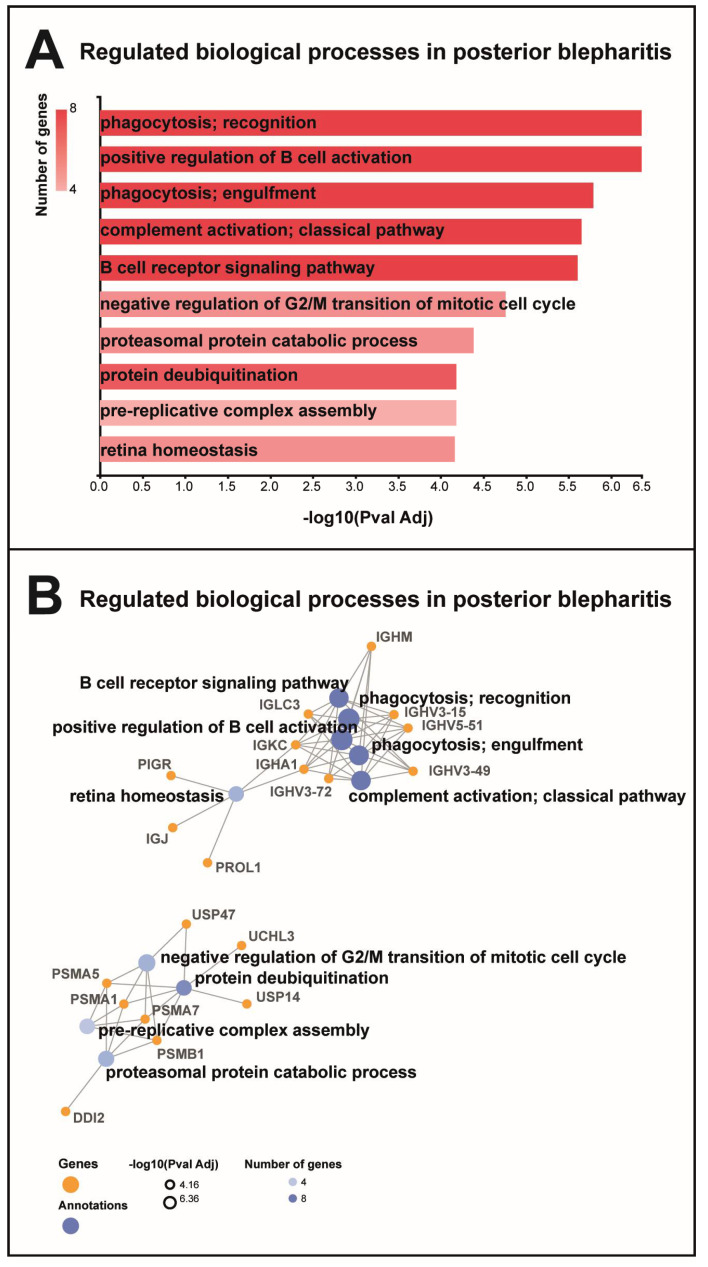
Gene ontology biological processes. (**A**) MGD was associated with regulation of proteins involved in B cell activation, phagocytosis, and complement activation. MGD was also associated with the regulation of proteins involved in mitotic cell cycle, protein deubiquitination, and proteasomal catabolic processes. (**B**) Proteins involved in B cell activation, phagocytosis, and complement activation were immunoglobulin components, including immunoglobulin heavy variable 5–51 (IGHV5-51), Ig mu chain C region (IGHM), immunoglobulin lambda constant 3 (IGLC3), immunoglobulin heavy variable 3–15 (IGHV3-15), Ig alpha-1 chain C region (IGHA1), immunoglobulin heavy variable 3–72 (IGHV3-72), immunoglobulin heavy variable 3–49 (IGHV3-49), and Ig kappa chain C region (IGKC). Proteins involved in proteasomal catabolic processes included proteasomal subunits alpha types 1, 5, and 7 and subunit beta type 1 (PSMA1, PSMA5, PSMA7, and PSMB1). Proteins involved in deubiquitination included proteasomal subunits (PSMA1, PSMA5, PSMA7, and PSMB1) and ubiquitin carboxyl-terminal hydrolases 14 (USP14), 47 (USP47), isozyme L3 (UCHL3), and protein DDI1 homolog 2 (DDI1).

**Figure 5 ijms-24-15115-f005:**
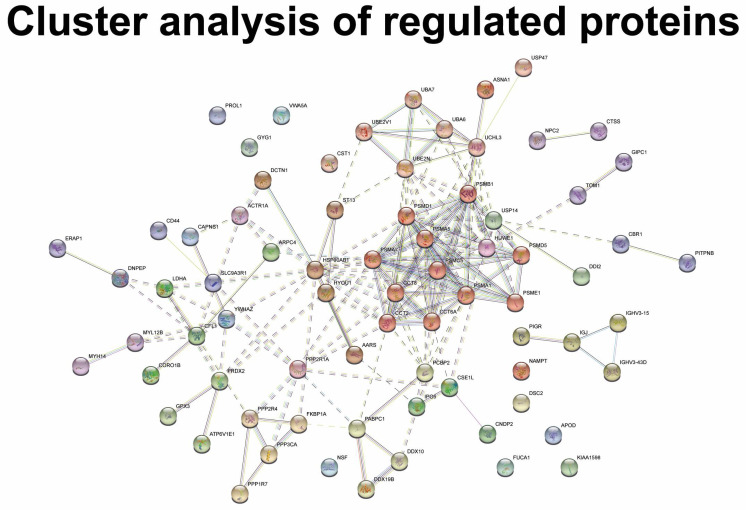
STRING cluster analysis of differentially regulated proteins in MGD. A number of clusters were identified. A major cluster consisted of proteasomal subunits (PSMA1, PSMA5, PSMA7, PSMB1), 26S protease regulatory subunit 8 (PSMC5), proteasome activator complex subunit 1 (PSME1), T-complex protein 1 subunit theta (CCT8), T-complex protein 1 subunit zeta (CCT6A), and T-complex protein 1 subunit beta (CCT2). A second cluster was formed by ubiquitin-like modifier-activating enzyme 6 (UBA6), ubiquitin-like modifier-activating enzyme 7 (UBA7), ubiquitin-conjugating enzyme E2 variant 1 (UBE2V1), ubiquitin-conjugating enzyme E2 N (UBE2N), and ubiquitin carboxyl-terminal hydrolase isozyme L3 (UCHL3). A third cluster was formed by serine/threonine-protein phosphatase 2A activator (PPP2R4), protein phosphatase 1 regulatory subunit 7 (PPP1R7), serine/threonine-protein phosphatase 2B catalytic subunit alpha isoform (PPP3CA), and peptidyl-prolyl cis-trans isomerase FKBP1A (FKBP1A). Another cluster was formed by immunoglobulin chains including immunoglobulin J chain (IGJ), immunoglobulin heavy variable 3–15 (IGHV3-15), Ig heavy chain V-III region DOB (IGHV3-43D), and polymeric immunoglobulin receptor (PIGR). Another cluster was formed by ATP-dependent RNA helicase DDX19B (DDX19B), probable ATP-dependent RNA helicase DDX10 (DDX10), polyadenylate-binding protein 1 (PABPC1), and poly(rC)-binding protein 2 (PCBP2).

**Table 1 ijms-24-15115-t001:** Significantly regulated proteins ordered according to fold change.

Protein ID	Protein Name	Gene Name	*p*-Value	MGD/Control
P11940	Polyadenylate-binding protein 1	*PABPC1*	0.026	2.67
Q14203-3	Dynactin subunit 1	*DCTN1*	0.022	2.50
P55060-3	Exportin-2	*CSE1L*	0.038	2.27
Q99460	26S proteasome non-ATPase regulatory subunit 1	*PSMD1*	0.037	1.92
P43490	Nicotinamide phosphoribosyltransferase	*NAMPT*	0.027	1.91
O43681	ATPase ASNA1	*ASNA1*	0.030	1.86
A0AVT1	Ubiquitin-like modifier-activating enzyme 6	*UBA6*	0.010	1.81
P41226	Ubiquitin-like modifier-activating enzyme 7	*UBA7*	0.017	1.78
P61088	Ubiquitin-conjugating enzyme E2 N	*UBE2N*	0.002	1.76
Q16401	26S proteasome non-ATPase regulatory subunit 5	*PSMD5*	0.030	1.72
Q15435	Protein phosphatase 1 regulatory subunit 7	*PPP1R7*	0.029	1.70
P30153	Serine/threonine-protein phosphatase 2A 65 kDa regulatory subunit A alpha isoform	*PPP2R1A*	0.018	1.68
P48739	Phosphatidylinositol transfer protein beta isoform	*PITPNB*	0.019	1.65
Q7Z6Z7-2	E3 ubiquitin-protein ligase HUWE1	*HUWE1*	0.005	1.65
P49588	Alanine--tRNA ligase, cytoplasmic	*AARS*	0.022	1.63
Q15257-3	Serine/threonine-protein phosphatase 2A activator	*PPP2R4*	0.040	1.59
O60784	Target of Myb protein 1	*TOM1*	0.045	1.59
O14908	PDZ domain-containing protein GIPC1	*GIPC1*	0.021	1.58
P62942	Peptidyl-prolyl cis-trans isomerase FKBP1A	*FKBP1A*	0.021	1.57
P25786	Proteasome subunit alpha type-1	*PSMA1*	0.007	1.57
P78371	T-complex protein 1 subunit beta	*CCT2*	0.045	1.54
P20618	Proteasome subunit beta type-1	*PSMB1*	0.038	1.54
P50990	T-complex protein 1 subunit theta	*CCT8*	0.033	1.54
P46459	Vesicle-fusing ATPase	*NSF*	0.047	1.54
O00410	Importin-5	*IPO5*	0.028	1.52
P61163	Alpha-centractin	*ACTR1A*	0.028	1.50
Q9ULA0-2	Aspartyl aminopeptidase	*DNPEP*	0.043	1.49
P36543	V-type proton ATPase subunit E 1	*ATP6V1E1*	0.022	1.48
P15374	Ubiquitin carboxyl-terminal hydrolase isozyme L3	*UCHL3*	0.015	1.46
P28066	Proteasome subunit alpha type-5	*PSMA5*	0.043	1.46
Q08209-3	Serine/threonine-protein phosphatase 2B catalytic subunit alpha isoform	*PPP3CA*	0.030	1.44
P59998	Actin-related protein 2/3 complex subunit 4	*ARPC4*	0.028	1.40
Q13404	Ubiquitin-conjugating enzyme E2 variant 1	*UBE2V1*	0.028	1.40
Q96K76-2	Ubiquitin carboxyl-terminal hydrolase 47	*USP47*	0.038	1.39
P54578	Ubiquitin carboxyl-terminal hydrolase 14	*USP14*	0.008	1.39
Q9BR76	Coronin-1B	*CORO1B*	0.015	1.38
P32119	Peroxiredoxin-2	*PRDX2*	0.045	1.38
P62195	26S protease regulatory subunit 8	*PSMC5*	0.049	1.38
O00534	von Willebrand factor A domain-containing protein 5A	*VWA5A*	0.043	1.36
O14950	Myosin regulatory light chain 12B	*MYL12B*	0.039	1.36
Q15366-7	Poly(rC)-binding protein 2	*PCBP2*	0.047	1.35
P08238	Heat shock protein HSP 90-beta	*HSP90AB1*	0.033	1.33
P46976-2	Glycogenin-1	*GYG1*	0.029	1.29
P04632	Calpain small subunit 1	*CAPNS1*	0.042	1.28
Q5TDH0-3	Protein DDI1 homolog 2	*DDI2*	0.033	1.27
P63104	14-3-3 protein zeta/delta	*YWHAZ*	0.035	1.27
Q9UMR2-2	ATP-dependent RNA helicase DDX19B	*DDX19B*	0.016	1.27
A0MZ66-5	Shootin-1	*KIAA1598*	0.018	1.27
P00338	L-lactate dehydrogenase A chain	*LDHA*	0.037	1.26
P16152	Carbonyl reductase [NADPH] 1	*CBR1*	0.012	1.26
P40227	T-complex protein 1 subunit zeta	*CCT6A*	0.033	1.25
Q96KP4	Cytosolic non-specific dipeptidase	*CNDP2*	0.025	1.24
O14745	Na(+)/H(+) exchange regulatory cofactor NHE-RF1	*SLC9A3R1*	0.037	1.24
O14818	Proteasome subunit alpha type-7	*PSMA7*	0.045	1.24
P23528	Cofilin-1	*CFL1*	0.019	1.23
Q7Z406-6	Myosin-14	*MYH14*	0.035	1.23
P50502	Hsc70-interacting protein	*ST13*	0.032	1.22
Q06323	Proteasome activator complex subunit 1	*PSME1*	0.013	1.22
Q9NZ08	Endoplasmic reticulum aminopeptidase 1	*ERAP1*	0.037	0.69
P0DOY3	Immunoglobulin lambda constant3	*IGLC3*	0.049	0.69
P01834	Ig kappa chain C region	*IGKC*	0.031	0.67
P04066	Tissue alpha-L-fucosidase	*FUCA1*	0.017	0.66
P22352	Glutathione peroxidase 3	*GPX3*	0.045	0.66
P01619	Ig kappa chain V-III region B6	*IGKV3-20*	0.038	0.65
P0DOX7	Immunoglobulin kappa light chain	*n/a*	0.013	0.65
Q9Y4L1	Hypoxia up-regulated protein 1	*HYOU1*	0.022	0.65
P16070	CD44 antigen	*CD44*	0.031	0.63
P61916-2	Epididymal secretory protein E1	*NPC2*	0.019	0.61
P01833	Polymeric immunoglobulin receptor	*PIGR*	0.032	0.59
A0A0C4DH68	Immunoglobulin kappa variable 2-24	*IGKV2-24*	0.017	0.59
P01876	Ig alpha-1 chain C region	*IGHA1*	0.014	0.58
A0A0B4J1Y9	Immunoglobulin heavy variable 3-72	*IGHV3-72*	0.031	0.58
P01591	Immunoglobulin J chain	*IGJ*	0.023	0.57
A0A0C4DH69	Immunoglobulin kappa variable 1-9	*IGKV1-9*	0.004	0.55
P25774-2	Cathepsin S	*CTSS*	0.0001	0.55
P01594	Ig kappa chain V-I region AU	*IGKV1-33*	0.010	0.54
Q99935	Proline-rich protein 1	*PROL1*	0.040	0.54
A0A0C4DH38	Immunoglobulin heavy variable 5-51	*IGHV5-51*	0.013	0.54
Q13206	Probable ATP-dependent RNA helicase DDX10	*DDX10*	0.047	0.53
A0A0B4J1V0	Immunoglobulin heavy variable 3-15	*IGHV3-15*	0.011	0.53
A0A0A0MS15	Immunoglobulin heavy variable 3-49	*IGHV3-49*	0.003	0.51
P01700	Ig lambda chain V-I region HA	*IGLV1-47*	0.010	0.50
Q02487-2	Desmocollin-2	*DSC2*	0.018	0.49
P04430	Ig kappa chain V-I region BAN	*IGKV1-16*	0.0020	0.48
P0DP04	Ig heavy chain V-III region DOB	*IGHV3-43D*	0.0080	0.47
P01871	Ig mu chain C region	*IGHM*	0.0020	0.46
P05090	Apolipoprotein D	*APOD*	0.0040	0.30
P01037	Cystatin-SN	*CST1*	0.0080	0.23

**Table 2 ijms-24-15115-t002:** Proteins with correlation with NBUT.

ProteinID	Protein Name	*p*-Value	Correlation
O14950	Myosin regulatory light chain 12B	0.0022	0.84
Q96K76-2	Ubiquitin carboxyl-terminal hydrolase 47	0.0263	0.77
P50502	Hsc70-interacting protein	0.035	0.67
Q15366-7	Poly(rC)-binding protein 2	0.036	0.66
Q9NZ08	Endoplasmic reticulum aminopeptidase 1	0.043	−0.68
A0A0B4J1V0	Immunoglobulin heavy variable 3–15	0.018	−0.76
P01871	Ig mu chain C region	0.0098	−0.77

**Table 3 ijms-24-15115-t003:** Samples for proteomics analysis.

	MGD	Controls	*p*-Value
**Number of samples (n)**	10	11	
**Age (years)**	63.8 ± 6.30	63.2 ± 5.7	0.82
**Sex (F/M)**	5/5	5/6	
**Meiboscore (grade)**	1.4 ± 0.5	0.0 ± 0.0	<0.0001
**Schirmer (mm)**	13.4 ± 1.3	17.3 ± 2.2	0.0001
**OSDI (score)**	38.2 ± 8.0	1.5 ± 1.8	<0.0001
**Tear meniscus (mm)**	0.37 ± 0.07	0.36 ± 0.04	0.63
**NBUT (s)**	10.6 ± 2.4	14.2 ± 1.8	0.00087

Data are expressed as n or mean ± standard deviation.

## Data Availability

Data resulting from the database search are available in the Appendix A.

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
