# Peer review of "Meibomian Gland Dysfunction Is Associated with Low Levels of Immunoglobulin Chains and Cystatin-SN"

_ijms, 2023, doi:10.3390/ijms242015115_

Round 1
Reviewer 1 Report
While the number of patients is very low, this study brings valuable information on the proteome of a patient with MGD and most importantly on the biochemical pathways involved. I believe it is of value to the readers of the Journal, after some clarifications:
1. Paragraph starting with 59 - while the state of knowledge on MGD proteome is limited, I suggest a short review of some known differently expressed proteins in such cases.
2. Line 232 - please mention details on meiboscore and MG drop out% - was a meibography performed?
3. Line 88 and after - were apolipoprotein D, Ig mu chain C, polyadenylate-binding protein 1, dynactin subunit 1 and exportin-2 correlated with clinical variables (Schirmer, NBUT, etc.)?
4. Why is Materials and Methods after Results and Discussion?
5. Please update References for more recent ones (3, 20, 12, 9)
6. Line 187 - please rephrase: To the best of our knowledge there is only limited knowledge about the role of immunoglobulins in blepharitis, but our data indicate that immunoglobulins[...]
Reviewer 2 Report
Interestin study with a small number of participants, considering the large number of cases with dry eye syndrome.
Good results .
Discusion need improve.
they dont have a section a conclusions of the study
Author Response
Please see the attachement
